# Fermentation Quality and In Vitro Digestibility of Sweet Corn Processing Byproducts Silage Mixed with Millet Hull or Wheat Bran and Inoculated with a Lactic Acid Bacteria

Meng Yu [†], Peng Wang *,[†], Fuhou Li [ID], Jiarui Du, Yitong Jin, Tianyue Zhao, Qixuan Yi, Hongyu Tang and Bao Yuan *[ID]

College of Animal Sciences, Jilin University, Changchun 130062, China; yumeng22@mails.jlu.edu.cn (M.Y.); lifh@jlu.edu.cn (F.L.); dujr23@mails.jlu.edu.cn (J.D.); ytjin23@mails.jlu.edu.cn (Y.J.); tianyue22@mails.jlu.edu.cn (T.Z.); yiqx21@mails.jlu.edu.cn (Q.Y.); tanghy@jlu.edu.cn (H.T.)
* Correspondence: pengwang@jlu.edu.cn (P.W.); yuan_bao@jlu.edu.cn (B.Y.)
[†] These authors contributed equally to this work.

**Abstract:** The aim of the experiment was to investigate the effect of different ratios of excipient (millet hull or wheat bran) and LAB inoculation on the fermentation quality and in vitro digestibility of a mixed silage of SCPBs. The preliminary experimental results showed that inoculating with lactic acid bacteria (LAB) directly in the fresh sweet corn processing byproduct (SCPBs) silage had a higher ammonia nitrogen/total nitrogen (AN/TN) ratio and lower silage fermentation quality due to high moisture content. Subsequently, millet hull or wheat bran were mixed with SCPBs in a 7:3 ($T_1$), 8:2 ($T_2$), and 9:1 ($T_3$) ratio and ensiled with LAB. Under the condition of each mixing ratio, the silage treatments were categorized into groups without any additives (control) and with LAB. Fermentation quality, in vitro digestibility, chemical composition, and energy values were determined after 45 days of silage. The pH, AN/TN, neutral detergent fiber, acid detergent fiber, and acid detergent lignin were lowest in the SCPBs and millet hull mixed silage (SMH) group under the $T_3$ treatment, whereas they were lowest in the SCPBs and wheat bran mixed silage (SWB) group under the $T_2$ treatment. The mean lactic acid and acetic acid values were higher in the SWB group than in the SMH group (6.92, 6.81 vs. 4.00, 4.52). Under the $T_3$ treatment in the SMH group, AN/TN was significantly reduced with the addition of LAB (4.52 vs. 4.37, $p < 0.05$). The SMH group had the highest crude protein (CP) under the $T_3$ treatment, whereas the SWB group had the highest CP under the $T_2$ treatment. The mean CP in the SWB group was higher than that of the SMH group (18.17, 19.44 vs. 10.55, 10.55). Under the $T_1$ treatment, in the SWB group, the addition of LAB resulted in a significant increase in in vitro crude protein digestibility ($p < 0.05$). The results showed that silage fermentation quality and in vitro digestibilitv55y improved with the addition of LAB. The optimum mixing ratio for the SWB group was 9:1 and 8:2 for the SMH group.

**Keywords:** sweet corn processing byproducts; mixed silage; lactic acid bacteria; fermentation quality; in vitro digestibility; wheat bran; millet hull

## 1. Introduction

With the transition of animal husbandry toward intensive and large-scale farming, the demand for feed is also increasing. However, restrictions on raw feed materials and rising prices have hindered the production of ruminants [1]. Therefore, unconventional roughage resources are receiving increasing attention in the livestock industry.

Sweet corn, also known as fruit corn, is native to the American continent and has a high sugar content and low starch content [2]. With the increasing demand for sweet corn products, many byproducts from sweet corn processing will also be generated. As reported in a previous study, sweet corn processing byproducts (SCPBs) account for up to 20–30% of the total corn output [3]. The byproducts mainly include corn husks, corn cobs, and

broken corn kernels. If the SCPBs could be systematically utilized, it could significantly reduce resource waste and environmental pollution. In addition, it can also alleviate the pressure on feed resources caused by insufficient conventional feed. However, an SCPBs are characterized by high soluble carbohydrate and moisture contents, making it difficult to preserve for a long period.

After anaerobic fermentation, the silage material has significant advantages such as a sweet smell, soft and juicy, good palatability, etc. Meanwhile, it can better preserve the nutrients in the raw material, such as proteins, vitamins, and minerals [4]. In addition to sufficient soluble sugars, appropriate moisture is also a necessary consideration for preparing high-quality silage [5]. Thus, some excipients with low moisture content should also be considered in SCPBs silage preparation. Both millet hull and wheat bran are common agricultural byproducts of grain processing, and their annual production is considerable, and they are mainly used as raw animal feed materials [6]. In some high-moisture silage, 10% wheat bran is often added to reduce the moisture content of silage, thereby improving fermentation quality [7]. However, there are few studies on millet hull as a feed ingredient. Numerous studies have shown that the nutrient content of millet hull is comparable to that of rice hulls [8]. In addition, rice hulls are often used as moisture regulators in silage production, e.g., adding 10% rice hull can significantly improve the fermentation quality of high-moisture Chinese sugarcane silage. Thus, we hypothesized that mixing different proportions of millet hull or wheat bran with SCPBs in silage production can result not only in the regulation of moisture content but also effectively improve fermentation quality and the utilization of grain processing byproducts.

Lactic acid bacteria (LAB) are widely used to ensure silage quality by improving the silage fermentation process and aerobic silage stability. In the early stage of fermentation, the addition of LAB can significantly increase the number of LAB in the silage so that the silage quickly enters the lactic acid (LA) fermentation stage, and the pH decreases rapidly [9]. The LAB species commonly used in silage production include *Lactobacillus*, *Enterococcus*, *Lactococcus*, *Pediococcus*, and *Bacillus* [10]. In this experiment, we added LAB to the SCPBs silage to ensure successful silage fermentation.

Therefore, the aim of this study was to investigate the effect of LAB and different proportions of the excipient on the fermentation quality and in vitro digestibility of silage mixed with SCPBs.

## 2. Materials and Methods

*2.1. Silage Preparation*

2.1.1. Step 1, (Preliminary Experiment) SCPBs Silage

To compare the effects of SCPBs silage alone and mixed silage, we first conducted a preliminary experiment. The experiment was conducted at the Animal Nutrition and Experimental Base, College of Animal Science, Jilin University, Changchun, Jilin Province, China (43.88° N, 125.35° E, elevation 300 m). SCPBs were obtained from Jilin Xinyu Beixian Rice Technology Co., Ltd, Changchun, China. The LAB strain used in this study was *Lactobacillus plantarum* LP1, which was purchased from SNOW BRAND SEED Co., Ltd. of Sapporo, Japan. The preliminary experiment included the following two treatments: (1) control and (2) LAB (the inoculation rate was $5 \times 10^6$ CFU $g^{-1}$ fresh weight). A total of 600 g of each chopped SCPBs were vacuum sealed in 140 mm × 200 mm polythene bags (purchased from Winstable Co., Ltd, Shanghai, China) with 5 replicates for each treatment. For the LAB treatment, the LAB was dissolved in distilled water following the instructions, sprayed evenly over the samples using a mini-sprayer and mixed thoroughly. The same dose of distilled water was added to the control group. A total of 10 samples were prepared and analyzed after 45 days of ensiling for fermentation quality, chemical composition, and in vitro digestibility.

2.1.2. Step 2, Mixed Silage Preparation of SCPBs with Millet Hull or Wheat Bran

Millet hull and wheat bran were obtained from Jilin Dabeinong Agricultural and Animal Husbandry Science and Technology Co., Ltd, Changchun, China. The experiment consisted of two treatment groups as follows: (1) SCPBs and millet hull mixed silage (SMH) group; and the (2) SCPBs and wheat bran mixed silage (SWB) group. Ground millet hull or wheat bran were mixed with SCPBs at ratios of 7:3 ($T_1$), 8:2 ($T_2$), and 9:1 ($T_3$). The final moisture content in the three ratios of each mixed treatment was 55%, 60%, and 65%. There were six groups in total, with six replicates in each group. The LAB strain, method of addition, and number of viable bacteria used were consistent with those used in the preliminary experiment. Fermentation quality, chemical composition, and in vitro digestibility were determined after 45 days of ensiling. After mixing, the samples were placed in 5 L plastic buckets at a density of $780 \pm 35$ kg m$^{-3}$ and stored at 20–25 °C at room temperature.

*2.2. Chemical Composition Analysis*

The fresh raw material and silage samples were placed in a blast drying oven at 60 °C for 72 h, at which time the dry matter content remained stable. The dried samples were pulverized using a crusher (2500C, Yongkang Red Sun Electromechanical Co., Ltd., Yongkang City, China), passed through a 1 mm sieve and then stored in polythene bags. Dry matter (DM), organic matter (OM), and crude protein (CP) were analyzed according to the methods of the Association of Official Analytical Chemists (AOAC) [11]. Neutral detergent fiber (NDF), acid detergent fiber (ADF), and acid detergent lignin (ADL) contents were determined using a fiber analyzer (SLQ-6A, Shanghai Fiber Testing Instrument Co., Ltd., Shanghai, China) according to Van Soest et al. [12]. Water-soluble carbohydrate (WSC) levels were determined by the sulfuric acid–anthrone method. The buffering capacity (BC) was measured using the method of Playne and McDonald [13]. Gross energy (GE) was determined using an oxygen bomb calorimeter (HXHW-5000, Hengxin Instrumentation Co., Ltd., Shanghai, China).

*2.3. Fermentation Quality and Microbiological Analyses*

Twenty grams of the silage sample was mixed with 180 mL of distilled water, stirred well, placed in a refrigerator at 4 °C overnight, and finally filtered through quantitative filter paper [14]. The pH was measured using a pH meter (FiveGo; Mettler Toledo, Greifensee, Switzerland). Ammonia nitrogen (AN) was measured by water vapor distillation, after which the ratio of AN to total nitrogen (AN/TN) was calculated [15]. Lactic acid (LA), acetic acid (AA), propionic acid (PA), and butyric acid (BA) were measured via high-performance liquid chromatography (column: Shodex RS Pak KC-811, Showa Denko KK, Kawasaki, Japan; detector: DAD, 210 nm, SPD-20A, Shimadzu Co., Ltd., Kyoto, Japan).

Under aseptic conditions, 20 g of fresh sample was taken into a glass triangular vial containing 180 mL of sterile physiological saline (0.85% NaCl). After sealing, they were shaken on a shaker for 30 min. They were then sequentially diluted with sterile physiological saline in a $10^{-1}$–$10^{-7}$ gradient. Three optimal dilutions were selected for the determination of the number of colonies ($10^{-4}$–$10^{-6}$). A total of 1 mL of the selected dilution of bacterial solution was aspirated and spread evenly on the agar medium. Three Petri dishes were prepared for each dilution. The lactic acid bacteria were counted on De Man, Rogosa and Sharp agar media (BKMAM Biotechnology Co., Ltd., Changsha, China) and incubated for 72 h at 30 °C, under anaerobic conditions. Yeasts and molds were counted on potato dextrose agar medium (BKMAM Biotechnology Co., Ltd., Changsha, China) and incubated for 96 h at 30 °C. Microorganisms were counted on Petri dishes of 20–200 cfu. All microbiological data were then converted to $\log_{10}$ CFUs g$^{-1}$ and the results were recorded as fresh weight [16].

### 2.4. In Vitro Digestibility Analysis

The study was conducted in strict compliance with the "Guidelines for Ethical Review of Laboratory Animal Welfare in China" and was approved by the Ethics Committee for Laboratory Animal Welfare of Jilin University (License No.: SY202009600). A total of 0.5 g of dried sample was placed in a filter bag (ANKOM F57; aperture 25 μm; ANKOM Technology Corporation; Macedon, NY, USA), sealed with a sealer, and stored in a sealed 130 mL in vitro digestion culture tube. Four F1-generation German meat Merinos crossed with lesser-tailed frosted sheep, fitted with permanent fistulas, were selected (fed alfalfa and corn silage twice daily; DM: 88.33% FW, CP: 11.76% DM, NDF: 38.42% DM, ADF: 17.88% DM). Rumen fluid was collected 1 h before the morning feeding, filtered through 4 layers of gauze and placed in a prewarmed thermos flask at 39 °C with continuous $CO_2$ exposure. The mixed buffer solution (prepared using the Longland et al. test method [17]) and rumen fluid were prepared as a mixed culture solution in a ratio of 1:4. Then, 50 mL of the mixed culture was put it in a culture tube and shaken together with the sample in a 39 °C water bath for 48 h. The filter bag was removed from the culture bottle after cultivation and rinsed with cold distilled water. The filter bags were then dried in a forced-air oven (100 °C, 24 h), and the residue was weighed, analyzed, and used to calculate the in vitro dry matter digestibility (*IV*DMD), in vitro organic matter digestibility (*IV*OMD), in vitro crude protein digestibility (*IV*CPD), and in vitro neutral detergent fiber digestibility (*IV*NDFD) [18]. In vitro nutrient digestibility = (content of nutrients before in vitro digestion − content of nutrients after in vitro digestion)/content of nutrients before in vitro digestion × 100%.

### 2.5. Statistical Analysis

The data were analyzed using SPSS (version 26; International Business Machines Corporation; Armonk, NY, USA). The preliminary experiment comprised an independent samples *t*-test to analyze whether there were significant differences in the mean values of fermentation quality, chemical composition, individual energy values, and in vitro digestibility between the control and LAB groups.

A general linear model of SPSS was used with the excipients, the ratios of excipients and SCPBs, and the inoculant as fixed factors to assess the effect of each factor on fermentation quality, chemical composition, energy and in vitro digestibility, and their interaction effects.

## 3. Results

### 3.1. Chemical Composition of Raw Materials before Ensiling

The characteristics of SCPBs, wheat bran, and millet hull before silage fermentation are shown in Table 1. The moisture content of the SCPBs is noted to reach 75.45%. The BC value of the SCPBs is 191.17 mEqkg$^{-1}$ DM, while the BC value of the millet hull is approximately 1.6 times higher than that of the wheat bran. The CP content of the SCPBs is 11.16% DM. The CP and WSC contents of wheat bran are approximately four times and two times greater than those of millet hull, respectively. The number of lactic acid bacteria attached to the SCPBs is 6.06 log10 CFUs g$^{-1}$ fresh weight and the amount of yeast is 3.35 log10 CFUs g$^{-1}$ fresh weight. No molds were detected in the SCPBs. Lactic acid bacteria, yeasts, and molds were not detected in wheat bran and millet hull.

**Table 1.** Characteristics of sweet corn processing byproducts, wheat bran, and millet hull.

| Item | SCPBs | Millet Hull | Wheat Bran |
|---|---|---|---|
| Chemical composition and buffering capacity | | | |
| Dry matter (%FW) | 24.55 | 93.05 | 88.76 |
| Organic matter (%DM) | 97.52 | 91.40 | 93.86 |
| Crude protein (%DM) | 11.16 | 4.56 | 17.89 |
| Neutral detergent fibre (%DM) | 75.76 | 73.86 | 56.28 |
| Acid detergent fibre (%DM) | 28.26 | 46.11 | 15.04 |
| Acid detergent lignin (%DM) | 4.50 | 27.65 | 6.94 |

**Table 1.** *Cont.*

| Item | SCPBs | Millet Hull | Wheat Bran |
|---|---|---|---|
| Water-soluble carbohydrate (%DM) | 6.06 | 3.25 | 6.77 |
| Buffering capacity (mEq kg$^{-1}$ DM) | 191.17 | 178.26 | 113.30 |
| Energy | | | |
| Gross energy (MJ kg$^{-1}$ DM) | 19.98 | 18.73 | 19.27 |
| Microbial counts | | | |
| Lactic acid bacteria (log$_{10}$ cfu g$^{-1}$ FW) | 6.06 | ND | ND |
| Yeasts (log$_{10}$ cfu g$^{-1}$ FW) | 3.35 | ND | ND |
| Moulds (log$_{10}$ cfu g$^{-1}$ FW) | ND | ND | ND |

SCPBs, sweet corn processing byproducts; FW, fresh weight; DM, dry matter; cfu, colony-forming units; ND, mean not detected.

### 3.2. Fermentation Characteristics of SCPBs Silage Alone

The fermentation quality of the fresh SCPBs silage is shown in Table 2. The AN/TN content is high in both the control and LAB groups. The chemical composition and in vitro nutrient digestibility of SCPBs silage are shown in Table 3. The CP in the LAB group is significantly greater than that of the control group ($p < 0.05$), and the DM in the control group is significantly lower than that of the LAB group ($p < 0.05$). All the indices of energy and in vitro digestibility are higher in the LAB groups than in the control groups, but no significant difference is observed.

**Table 2.** Fermentation quality in sweet corn processing byproduct silage prepared with lactic acid bacteria.

| Item ‡ | Additive † | | SEM | *p*-Value |
|---|---|---|---|---|
| | Control | LAB | | |
| pH | 3.54 | 3.56 | 0.007 | 0.067 |
| Ammonia nitrogen (%TN) | 8.07 | 7.84 | 0.321 | 0.508 |
| Lactic acid (%DM) | 17.84 | 19.01 | 1.506 | 0.481 |
| Acetic acid (%DM) | 8.21 | 8.78 | 0.650 | 0.471 |
| Propionic acid (%DM) | 0.85 | 1.02 | 0.359 | 0.666 |
| Butyric acid (%DM) | 6.87 | 7.17 | 0.565 | 0.623 |

SEM, standard error of the mean. † LAB, lactic acid bacteria; ‡ DM, dry matter; AN, ammonia nitrogen; TN, total nitrogen.

**Table 3.** Chemical composition, energy, and in vitro digestibility in sweet corn processing byproduct silage prepared with lactic acid bacteria.

| Item ‡ | Additive † | | SEM | *p*-Value |
|---|---|---|---|---|
| | Control | LAB | | |
| Dry matter (%FW) | 23.54 | 24.55 | 0.100 | 0.001 |
| Organic matter (%DM) | 97.52 | 97.41 | 0.056 | 0.119 |
| Crude protein (%DM) | 11.26 | 11.83 | 0.162 | 0.025 |
| Neutral detergent fiber (%DM) | 69.46 | 71.13 | 0.708 | 0.078 |
| Acid detergent fiber (%DM) | 32.20 | 31.39 | 1.423 | 0.623 |
| Acid detergent lignin (%DM) | 6.42 | 6.40 | 0.560 | 0.973 |
| Gross energy (MJ kg$^{-1}$ DM) | 21.34 | 21.77 | 0.171 | 0.066 |
| *IV*DMD (%DM) | 61.01 | 61.60 | 1.074 | 0.640 |
| *IV*OMD (%DM) | 63.46 | 64.07 | 1.039 | 0.615 |
| *IV*CPD (%DM) | 53.43 | 54.13 | 0.469 | 0.268 |
| *IV*GED (%DM) | 63.67 | 64.25 | 0.978 | 0.615 |
| *IV*NDFD (%DM) | 60.10 | 61.55 | 0.599 | 0.073 |

SEM, standard error of mean. † LAB, lactic acid bacteria. ‡ DM, dry matter; FW, fresh weight; *IV*DMD, in vitro dry matter digestibility; *IV*OMD, in vitro organic matter digestibility; *IV*CPD, in vitro crude protein digestibility; *IV*GED, in vitro gross energy digestibility; *IV*NDFD, in vitro neutral detergent fiber digestibility.

### 3.3. Step 2 Mixed Silage of the SCPBs with Wheat Bran or Millet Hull

#### 3.3.1. Fermentation Characteristics

The fermentation quality is shown in Table 4. There is an interaction effect of T $\times$ E on pH and AN/TN ($p < 0.001$). The SMH groups had the lowest pH and AN/TN under the $T_3$ treatment, while the SWB groups had the lowest pH and AN under the $T_2$ treatment. In the $T_3$ treatment of the SMH groups, the AN/TN ratio significantly decreased with the addition of LAB (4.52 vs. 4.37, $p < 0.05$). Moreover, E had a significant effect on LA and AA ($p < 0.001$). The mean values of LA and AA were higher in the SWB group than in the SMH group (6.92, 6.81 vs. 4.00, 4.52). In the SMH groups, the LA levels increased with increasing moisture content ($p < 0.05$).

**Table 4.** Fermentation quality in sweet corn processing byproducts mixed with millet hull or wheat bran for silage prepared with lactic acid bacteria.

| Item ‡ | T | Additives † | | | | Mean | SEM | Significance of Main Effects and Interactions | | | | | | |
|---|---|---|---|---|---|---|---|---|---|---|---|---|---|---|
| | | SMH | | SWB | | | | | | | | | | |
| | | Control | LAB | Control | LAB | | | T | A | E | T $\times$ A | T $\times$ E | A $\times$ E | T $\times$ A $\times$ E |
| pH | $T_1$ | 3.75 $^{aC}$ | 3.77 $^{aC}$ | 3.85 $^{bC}$ | 3.86 $^{bC}$ | 3.81 | 0.002 | <0.001 | 0.002 | <0.001 | 0.451 | <0.001 | 0.099 | 0.890 |
| | $T_2$ | 3.66 $^{aB}$ | 3.68 $^{bB}$ | 3.64 $^{aA}$ | 3.65 $^{aA}$ | 3.66 | | | | | | | | |
| | $T_3$ | 3.58 $^{aA}$ | 3.59 $^{aA}$ | 3.74 $^{bB}$ | 3.75 $^{bB}$ | 3.66 | | | | | | | | |
| | Average | 3.66 | 3.68 | 3.74 | 3.75 | | | | | | | | | |
| AN (%TN) | $T_1$ | 5.54 $^{dC}$ | 5.32 $^{cC}$ | 3.04 $^{bB}$ | 2.55 $^{aA}$ | 4.11 | 0.015 | <0.001 | <0.001 | <0.001 | 0.021 | <0.001 | 0.804 | 0.015 |
| | $T_2$ | 5.26 $^{bB}$ | 4.98 $^{bB}$ | 2.64 $^{aA}$ | 2.58 $^{aA}$ | 3.87 | | | | | | | | |
| | $T_3$ | 4.52 $^{cA}$ | 4.37 $^{bA}$ | 3.24 $^{aC}$ | 3.10 $^{aB}$ | 3.81 | | | | | | | | |
| | Average | 5.10 | 4.89 | 2.97 | 2.75 | | | | | | | | | |
| LA (%DM) | $T_1$ | 3.08 $^{a}$ | 3.82 $^{aA}$ | 6.67 $^{b}$ | 6.66 $^{b}$ | 5.06 | 0.086 | 0.001 | 0.672 | <0.001 | 0.147 | 0.171 | 0.303 | 0.572 |
| | $T_2$ | 4.42 $^{a}$ | 3.94 $^{aA}$ | 6.82 $^{b}$ | 6.48 $^{b}$ | 5.73 | | | | | | | | |
| | $T_3$ | 4.48 $^{a}$ | 4.98 $^{aB}$ | 7.26 $^{b}$ | 7.29 $^{b}$ | 5.69 | | | | | | | | |
| | Average | 4.00 | 4.25 | 6.92 | 6.81 | | | | | | | | | |
| AA (%DM) | $T_1$ | 1.26 | 1.56 | 1.72 | 1.89 | 1.61 | 0.038 | 0.587 | 0.288 | <0.001 | 0.079 | 0.366 | 0.070 | 0.431 |
| | $T_2$ | 1.51 | 1.42 | 1.85 | 1.60 | 1.59 | | | | | | | | |
| | $T_3$ | 1.01 $^{a}$ | 1.49 $^{b}$ | 1.83 $^{b}$ | 1.74 $^{b}$ | 1.52 | | | | | | | | |
| | Average | 1.26 | 1.49 | 1.80 | 1.74 | | | | | | | | | |
| PA (%DM) | $T_1$ | 0.64 $^{a}$ | 2.29 $^{bB}$ | 0.14 $^{a}$ | 0.16 $^{aB}$ | 0.81 | 0.010 | 0.164 | 0.829 | 0.507 | 0.191 | 0.222 | 0.155 | 0.586 |
| | $T_2$ | 0.76 | 0.68 $^{AB}$ | 2.23 | 0.11 $^{B}$ | 0.95 | | | | | | | | |
| | $T_3$ | 0.00 $^{a}$ | 0.00 $^{aA}$ | 0.02 $^{b}$ | 0.01 $^{aA}$ | 0.01 | | | | | | | | |
| | Average | 0.47 | 0.99 | 0.80 | 0.09 | | | | | | | | | |
| BA (%DM) | $T_1$ | 0.23 $^{aA}$ | 0.29 $^{aA}$ | 0.50 $^{b}$ | 0.30 $^{aA}$ | 0.33 | 0.210 | <0.001 | 0.143 | 0.010 | 0.115 | 0.019 | 0.013 | 0.016 |
| | $T_2$ | 0.42 $^{B}$ | 0.34 $^{A}$ | 0.41 | 0.39 $^{AB}$ | 0.39 | | | | | | | | |
| | $T_3$ | 0.46 $^{B}$ | 0.54 $^{B}$ | 0.51 | 0.49 $^{B}$ | 0.50 | | | | | | | | |
| | Average | 0.37 | 0.39 | 0.48 | 0.39 | | | | | | | | | |

$^{A–C}$ Means of the different ratios of excipient within a column with different superscripts differ in the same additive treatment ($p < 0.05$). $^{a–d}$ Means of additives treatments within a row with different superscripts differ in the same ratios of excipient ($p < 0.05$). SEM, standard error of mean; A, additive; E, excipient; T, different ratios of excipient mixed with SCPBs for silage. † LAB, lactic acid bacteria. SMH, sweet corn processing byproduct and millet hull mixed silage; SWB, sweet corn processing byproduct and wheat bran mixed silage. ‡ DM, dry matter; AN, ammonia nitrogen; TN, total nitrogen LA, lactic acid; AA, acetic acid; PA, propionic acid; BA, butyric acid.

#### 3.3.2. Chemical Composition of Mixed Silage

The chemical compositions of the SMH and SWB groups are shown in Table 5. T had a significant effect on DM and OM ($p < 0.001$). With the decreasing excipient concentration, the DM content of the silage also decreased, while the OM content increased. In the SMH group, the addition of LAB resulted in a significant increase in the OM content ($p < 0.05$). Although there was a T $\times$ E interaction effect on CP, the effect was too small to be of practical significance. The SMH group had the highest CP in the $T_3$ treatment, while the SWB group had the highest CP in the $T_2$ treatment. The mean CP was higher in the SWB group than in the SMH group (18.17, 19.44 vs. 10.55, 10.55). The interaction effect (T $\times$ E) was significant ($p < 0.001$) for NDF, ADF, and ADL. The SMH group had the lowest NDF, ADF, and ADL contents under the $T_3$ treatment, while the SWB group had the lowest NDF, ADF, and ADL contents under the $T_2$ treatment. The mean values of NDF, ADF and ADL

were lower in the SWB group than in the SMH group. In the SMH and SWB groups, the content of GE increased ($p < 0.001$) with decreasing excipient addition.

**Table 5.** Chemical composition and energy in sweet corn processing byproducts mixed with millet hull or wheat bran for silage prepared with lactic acid bacteria.

| Item ‡ | T | Additives † SMH Control | SMH LAB | SWB Control | SWB LAB | Mean | SEM | T | A | E | T × A | T × E | A × E | T × A × E |
|---|---|---|---|---|---|---|---|---|---|---|---|---|---|---|
| DM (%FW) | $T_1$ | 38.12 C | 38.01 C | 38.32 C | 38.18 C | 38.16 | | | | | | | | |
| | $T_2$ | 32.33 aB | 33.13 bB | 32.76 abB | 32.99 bB | 38.20 | 0.046 | <0.001 | 0.012 | 0.003 | 0.023 | 0.139 | 0.269 | 0.367 |
| | $T_3$ | 27.27 aA | 27.64 abA | 27.85 abA | 28.20 bA | 27.74 | | | | | | | | |
| | Average | 32.57 | 32.92 | 32.98 | 33.12 | | | | | | | | | |
| OM (%DM) | $T_1$ | 92.88 aA | 92.79 aA | 94.34 bA | 94.32 bA | 93.58 | | | | | | | | |
| | $T_2$ | 93.45 aB | 93.41 aB | 94.66 bB | 95.02 bB | 94.14 | 0.029 | <0.001 | 0.803 | <0.001 | 0.207 | <0.001 | 0.073 | 0.476 |
| | $T_3$ | 94.47 aC | 94.33 aC | 95.10 bC | 95.13 bB | 94.76 | | | | | | | | |
| | Average | 93.60 | 93.51 | 94.70 | 94.82 | | | | | | | | | |
| CP (%DM) | $T_1$ | 9.08 aA | 9.28 aA | 16.26 bA | 20.32 cC | 13.71 | | | | | | | | |
| | $T_2$ | 10.56 aB | 10.58 aB | 19.50 bC | 19.60 bB | 15.06 | 0.020 | <0.001 | <0.001 | <0.001 | <0.001 | <0.001 | <0.001 | <0.001 |
| | $T_3$ | 12.02 aC | 11.79 aC | 18.75 cB | 18.40 bA | 15.24 | | | | | | | | |
| | Average | 10.55 | 10.55 | 18.17 | 19.44 | | | | | | | | | |
| NDF (%DM) | $T_1$ | 79.17 bB | 80.09 bC | 57.02 a | 58.91 a | 38.80 | | | | | | | | |
| | $T_2$ | 78.43 bB | 78.37 bB | 56.78 a | 54.44 a | 67.00 | 0.218 | <0.001 | 0.403 | <0.001 | 0.029 | <0.001 | 0.150 | 0.207 |
| | $T_3$ | 74.56 bA | 74.53 bA | 58.30 a | 55.69 a | 65.77 | | | | | | | | |
| | Average | 77.39 | 77.66 | 57.37 | 56.34 | | | | | | | | | |
| ADF (%DM) | $T_1$ | 47.53 bB | 47.47 bB | 19.37 aA | 20.20 aA | 33.64 | | | | | | | | |
| | $T_2$ | 45.17 bB | 45.77 bB | 20.54 aA | 20.91 aA | 33.10 | 0.016 | 0.078 | 0.444 | <0.001 | 0.720 | <0.001 | 0.685 | 0.771 |
| | $T_3$ | 41.78 bA | 41.60 bA | 23.73 aB | 23.68 aB | 32.70 | | | | | | | | |
| | Average | 44.83 | 44.95 | 21.21 | 21.60 | | | | | | | | | |
| ADL (%DM) | $T_1$ | 20.95 bB | 21.14 bC | 6.81 a | 6.59 aA | 13.87 | | | | | | | | |
| | $T_2$ | 17.50 bA | 18.57 bB | 6.53 a | 6.85 aA | 12.36 | 0.123 | <0.001 | 0.426 | <0.001 | 0.011 | <0.001 | 0.337 | 0.020 |
| | $T_3$ | 17.83 cA | 15.25 bA | 7.66 a | 7.69 aB | 12.11 | | | | | | | | |
| | Average | 18.76 | 18.32 | 7.00 | 7.04 | | | | | | | | | |
| GE (MJ kg⁻¹ DM) | $T_1$ | 19.71 bA | 19.56 aA | 20.08 cA | 20.43 dA | 19.94 | | | | | | | | |
| | $T_2$ | 20.05 aA | 20.07 aB | 20.69 bB | 20.83 bB | 20.41 | 0.024 | <0.001 | 0.550 | <0.001 | 0.212 | 0.464 | 0.002 | 0.294 |
| | $T_3$ | 20.80 abB | 20.53 aC | 21.18 bC | 21.26 bC | 20.95 | | | | | | | | |
| | Average | 20.19 | 20.05 | 20.65 | 20.84 | | | | | | | | | |

A–C Means of different ratios of excipient within a column with different superscripts differ in the same additive treatment ($p < 0.05$). a–d Means of additives treatments within a row with different superscripts differ in the same ratios of excipient ($p < 0.05$). SEM, standard error of the mean; A, additive; E, excipient; T, different ratios of excipient mixed with SCPBs for silage. † LAB, lactic acid bacteria. SMH, sweet corn processing byproduct and millet hull mixed silage; SWB, sweet corn processing byproduct and wheat bran mixed silage. ‡ DM, dry matter; FW, fresh weight; OM, organic matter; CP, crude protein; NDF, neutral detergent fiber; ADF, acid detergent fiber; ADL, acid detergent lignin; GE, gross energy.

### 3.3.3. In Vitro Digestibility of Mixed Silage

The in vitro digestibility of the mixed silage is shown in Table 6. The interaction (T × E) had a significant effect on *IV*DMD, *IV*OMD, *IV*CPD, and *IV*NDFD ($p < 0.001$). The SMH group had the highest *IV*DMD, *IV*OMD, and *IV*CPD under the $T_3$ treatment, while the SWB group had the highest *IV*DMD, *IV*OMD, and *IV*CPD under the $T_2$ treatment. The *IV*NDFD was significantly higher in the SMH group under the $T_1$ and $T_2$ treatments than under the $T_3$ treatment ($p < 0.05$). In the SWB group, $T_1$, the addition of LAB increased the *IV*CPD.

**Table 6.** In vitro digestibility in sweet corn processing byproducts mixed with millet hull or wheat bran for silage prepared with lactic acid bacteria.

| Item ‡ | T | Additives † SMH Control | SMH LAB | SWB Control | SWB LAB | Mean | SEM | T | A | E | T × A | T × E | A × E | T × A × E |
|---|---|---|---|---|---|---|---|---|---|---|---|---|---|---|
| *IV*DMD (%DM) | $T_1$ | 46.51 aA | 46.32 aA | 66.87 b | 67.59 b | 56.82 | | | | | | | | |
| | $T_2$ | 48.89 aB | 48.55 aB | 67.56 b | 67.62 b | 58.15 | 0.473 | <0.001 | 0.964 | <0.001 | 0.715 | <0.001 | 0.235 | 0.862 |
| | $T_3$ | 52.36 aC | 52.11 aC | 66.20 b | 66.24 b | 59.23 | | | | | | | | |
| | Average | 49.25 | 48.99 | 66.88 | 67.15 | | | | | | | | | |

**Table 6.** *Cont.*

| Item ‡ | T | Additives † | | | | Mean | SEM | Significance of Main Effects and Interactions | | | | | | |
| | | SMH | | SWB | | | | | | | | | | |
| | | Control | LAB | Control | LAB | | | T | A | E | T × A | T × E | A × E | T × A × E |
| *IV*OMD (%DM) | T$_1$ | 50.14 $^{aA}$ | 50.09 $^{aA}$ | 71.10 $^{bB}$ | 71.39 $^{bB}$ | 60.68 | | | | | | | | |
| | T$_2$ | 52.32 $^{aB}$ | 51.92 $^{aB}$ | 71.23 $^{bB}$ | 71.17 $^{bB}$ | 61.66 | 0.115 | <0.001 | 0.826 | <0.001 | 0.825 | <0.001 | 0.579 | 0.971 |
| | T$_3$ | 55.54 $^{aC}$ | 55.45 $^{aC}$ | 69.32 $^{bA}$ | 69.33 $^{bA}$ | 62.41 | | | | | | | | |
| | Average | 52.67 | 52.49 | 70.55 | 70.63 | | | | | | | | | |
| *IV*CPD (%DM) | T$_1$ | 45.07 $^{a}$ | 45.92 $^{bA}$ | 56.49 $^{cA}$ | 60.07 $^{dC}$ | 51.89 | | | | | | | | |
| | T$_2$ | 46.57 $^{a}$ | 46.33 $^{aA}$ | 58.71 $^{bB}$ | 58.71 $^{bB}$ | 52.58 | 0.114 | <0.001 | <0.001 | <0.001 | <0.001 | <0.001 | 0.088 | <0.001 |
| | T$_3$ | 47.08 $^{a}$ | 47.70 $^{aB}$ | 56.14 $^{bA}$ | 55.56 $^{bA}$ | 51.62 | | | | | | | | |
| | Average | 46.24 | 46.65 | 57.11 | 58.11 | | | | | | | | | |
| *IV*NDFD (%DM) | T$_1$ | 73.75 $^{bB}$ | 74.64 $^{bC}$ | 52.38 $^{a}$ | 54.20 $^{a}$ | 63.74 | | | | | | | | |
| | T$_2$ | 73.03 $^{bB}$ | 72.98 $^{bB}$ | 52.14 $^{a}$ | 49.88 $^{a}$ | 62.01 | 0.107 | <0.001 | 0.402 | <0.001 | 0.029 | <0.001 | 0.151 | 0.208 |
| | T$_3$ | 69.30 $^{bA}$ | 69.27 $^{bA}$ | 53.61 $^{a}$ | 51.09 $^{a}$ | 60.82 | | | | | | | | |
| | Average | 72.03 | 72.30 | 52.71 | 51.72 | | | | | | | | | |

$^{A-C}$ Means of different ratios of excipient within a column with different superscripts differ in the same additive treatment ($p < 0.05$). $^{a-d}$ Means of additives treatments within a row with different superscripts differ in the same ratios of excipient ($p < 0.05$). SEM, standard error of mean; A, additive; E, excipient; T, different ratios of excipient mixed with SCPBs for silage. † LAB, lactic acid bacteria. SMH, sweet corn processing byproduct and millet hull mixed silage; SWB, sweet corn processing byproduct and wheat bran mixed silage. ‡ DM, dry matter; *IV*DMD, in vitro dry matter digestibility; *IV*OMD, in vitro organic matter digestibility; *IV*CPD, in vitro crude protein digestibility; *IV*NDFD, in vitro neutral detergent fiber digestibility.

## 4. Discussion

### 4.1. Effects of the Mixing Proportion and Lactic Acid Bacteria Addition on Silage Fermentation Quality

The moisture content is the main factor affecting silage fermentation. The fermentation quality of silage improved at 55–65% moisture content, and silage osmotic yield and nutrient losses decreased [19]. The WSC content is also closely related to the success of silage fermentation. As reported in a previous study, raw material with a moisture content of 70% should have a WSC content of 7–10% to ensure silage fermentation quality [20]. The moisture content of the SCPBs in this study reached 75.45%, and the WSC content was 6.06% DM. In a preliminary experiment, fresh SCPBs were directly ensiled and was found to have a relatively high AN/TN content (7.84–8.07% TN). Therefore, in this study, we prepared mixed silage using SCPBs and varying amounts of millet hull or wheat bran, two common excipients used in crop production. The aim was to improve the DM level and reduce the moisture content. It has been reported that when the moisture content of the raw material for corn silage is 60–70%, fermentation by LAB is more favorable [21]. Therefore, we adjusted three mixing ratios (T$_1$, T$_2$, and T$_3$) with moisture contents of 55% (T$_1$), 60% (T$_2$), and 65% (T$_3$), respectively. pH is an essential indicator for evaluating the quality of high moisture content silage [22]. When the pH is lower than 4.2, the growth of harmful microorganisms is inhibited, and the silage can be effectively preserved [23]. The pH of the SWB and SMH groups in this study varied from 3.64 to 3.86. The pH values of the T$_3$ SMH group and T$_2$ SWB group were the lowest. This indicates that better silage quality occurred in these groups. The AN/TN ratio reflects the level of protein and amino acid degradation in silage [24]. Compared with those in the T$_1$ and T$_2$ LAB groups in the SMH treatment, the AN/TN ratios were lower in the T$_3$ LAB group. This is probably due to the decrease in the moisture content caused by the change in the mixing ratios of millet hull and SCPBs. Because microorganisms exhibit protein hydrolysis activity in high moisture content silage, this condition encourages the growth and activity of microorganisms [25]. In contrast, the AN/TN content of the SWB treatment did not maintain this trend. This is likely the result of differences among the excipients. The addition of wheat bran to *Broussonetia papyrifera* silage reduced the AN/TN content according to Wang et al. [26]. Gül et al. reported that in caramba silage, the AN/TN ratio was lower in the 10% wheat bran group than in the 5% wheat bran group [27]. This finding is comparable with the results of the present study. This suggests that wheat bran can improve the fermentation quality of silage to a certain extent, probably because the addition of wheat bran reduces

the moisture content and inhibits the growth of heterozygous bacteria, thus promoting fermentation. In addition, the AN/TN ratios were greater in the SMH group than in the SWB group. A higher BC value was a major factor contributing to the lower fermentation quality of the SMH group (BC: 178.26 mEq kg $^{-1}$ DM). The high moisture content group in this experiment had a higher LA content and lower pH and PA values. A high moisture content easily leads to the growth of harmful bacteria, such as *Clostridium*, which produce BA. It has been proven that lowering the moisture content of silage could decrease BA production [28]. In this experiment, the moisture content and LA content decreased with the increasing millet hull ratio. This is in line with another study in which the LA and AA levels decreased in silage as the proportion of pistachio hulls increased [29]. Chen et al. [30] reported that a low moisture content is detrimental to LA accumulation. This can be attributed to the reduced activity of the bacteria in a low-moisture environment, resulting in a lower concentration of fermentation acid [31]. A lower moisture content inhibits the growth and multiplication of LAB, yet LAB are the most important microorganisms for successful silage fermentation. Vendramini et al. [32] found that, in the bermudagrass silage trials, lowering the moisture content led to a greater pH and lower LA and acetate contents. The LA content was higher in all SWB groups than in the SMH group. This is because the WSC content of wheat bran is 2.08 times greater than that of millet hull, and the use of wheat bran as an auxiliary supplement can increase the WSC content of silage. LAB multiply rapidly in silage with a high WSC content, resulting in a rapid decrease in pH, which inhibits the growth of undesirable bacteria [33]. High PA concentrations can result in the low fermentation efficiency in silage and facilitate secondary fermentation [34]. The PA content in good-quality silage should be less than 10 g kg $^{-1}$ DM [35]. The PA content of the silages in this experiment ranged from 0 to 2.29 g kg $^{-1}$ DM, which falls within the normal range for high-moisture silage fermentation. An excessive moisture content in the raw material or a lack of adequate WSC leads to Clostridial growth and BA production. The BA content was low (0.30–0.54% DM) in the groups in this study because the amount of the raw material WSC in this study was sufficient, and an appropriate moisture content was used.

LAB additives are mainly used to improve fermentation quality and prolong silage storage time by increasing LA production. Xia et al. [36] showed that the addition of LAB to high-moisture ryegrass silage resulted in a significant decrease in pH and a significant increase in DM and AA contents. Sun et al. [37] reported that the pH and AN levels were significantly lower but that the LA and AA levels were significantly greater in a test group than in a control group. However, there was no significant trend of pH reduction in the LAB group compared to the control group in this experiment. This is because the effect of the inoculant can differ based on the raw materials. For example, homofermentative LAB additives can reduce the pH of temperate and tropical forage and legume silages but have no effect on grass silages, such as corn, sorghum, and sugarcane silages [38]. As expected, compared with that of the control group, the AN/TN content in the $T_3$ SMH and $T_1$ SMH groups significantly decreased after LAB addition. LAB addition leads to a rapid reduction in the AN content in silage by reducing the level of protein breakdown and increasing DM recovery [39]. In addition, the AA content was greater in the LAB group than in the control group. Research has demonstrated that adding LAB to silage increases the content of LA and AA. After 56 days of ensiling, the LA content of the ryegrass silage inoculated with LAB was 110 g kg $^{-1}$ DM and the AA content was 43.3 g kg $^{-1}$ DM [40].

### 4.2. Effect of the Mixing Proportion and Lactic Acid Bacteria Additive on the Chemical Composition

In this experiment, the DM content of the SMH and SWB groups increased with increasing wheat bran and millet hull. Wan et al. [41] showed that with increasing DM levels in sudangrass silage at low moisture contents, the silage fermentation quality improved. In addition, in this experiment, the change in the mixing ratio had a significant effect on the CP concentration. The CP content of the SMH groups decreased with increasing millet

hull. This could be attributed to the lower CP content of millet hull. Moreover, plant cells remain active during the silage process, and the intracellular degradation enzymes degrade proteins to ammonia at low moisture contents [42]. This result was also demonstrated by Liu et al. [43], who reported that in fresh ryegrass silage trials, lowering the moisture content led to a decrease in the CP content. The findings of Zheng and Fitzgerald et al. [31,44] were also similar to the results of the present experiment. When NDF and ADF levels were low, the silage had greater nutritional value. As the moisture content decreased, the levels of ADF and ADL decreased in the SWB groups. This suggests that a low moisture content better preserves the nutrient content of mixed SWB silage. This can be explained by the fact that harmful microorganisms such as *C. difficile* and molds are inactive at low water contents and cause a low level of decomposition of the fibers and lignin in the feed. Kim et al. [9] observed an increasing trend in crude fiber content with decreasing round baled rye moisture content. Hashemzadeh-Cigari et al. [45] reported that wilting resulted in an increase in the DM content and a decrease in the NDF and ADF contents. This finding is similar to the results of the SWB groups in this study. The DM, WSC, and CP contents increased significantly, and NDF and ADF decreased with the addition of wheat bran. This indicates that the addition of wheat bran better preserved the nutrient content of the mixed silage. Interestingly, at high moisture content, the NDF, ADF, and ADL levels were lower in the SMH groups. This is the opposite of the trend observed in the SWB group. This may be related to the high fiber content of the raw millet hull. At low moisture levels, when a large amount of millet hull was mixed with the SCPBs, the fiber content increased. Aydin et al. reported that NDF and ADF increased with increasing almond shell content, which was attributed to the higher contents of ADF and NDF in almond shells [46].

In this study, the LAB addition did not have a significant effect on DM. Liu et al. [43] reported that the addition of *Lactobacillus plantarum* to Italian ryegrass silage had no significant effect on the dry matter content. Keady and Murphy's study also proved this point [47]. In the $T_1$ SWB group, the CP levels were significantly greater in the LAB group than in the control group. This indicates that adding LAB can preserve more CP without degradation. The results of the studies of Kennedy and Khota et al. [48,49] were similar to the results of this study. This is the result of improving silage fermentation quality to reduce protein degradation. Cao et al. [50] reported that the addition of *Lactobacillus* preparations to alfalfa silage increased the content of CP and organic acids other than BA and effectively reduced the numbers of Clostridia, molds, and yeasts. In the $T_3$ SMH group, the ADL level was lower in the LAB group than in the control group. The differences in NDF and ADF contents in the other LAB additive groups compared to those in the control group were not significant. Keady and Murphy reported that LAB addition reduced the NDF and ADF contents in ryegrass silage, but the effect was not significant [47]. These findings were similar to the results of the present study.

### 4.3. Effect of the Mixing Proportion and Lactic Acid Bacteria Additive on In Vitro Digestibility

The energy distribution potential of a feed can be assessed by measuring the content of *IV*DMD in the rumen of ruminants [51]. *IV*DMD can reflect the extent to which feed is degraded by rumen microbes in the rumen [52]. In general, the *IV*NDFD is one of the most important indicators for determining the nutritional value of silage. In this study, the *IV*DMD was greater in the $T_3$ SMH group than in the $T_1$ and $T_2$ SMH groups, whereas the ADF was lower in the $T_3$ SMH group than in the $T_1$ and $T_2$ SMH groups. ADF levels are negatively correlated with *IV*DMD [53]. The results of this experiment also verified this conclusion. In the SMH group, the *IV*DMD increased with increasing moisture content, while the opposite was true for the *IV*NDFD. *IV*DMD and *IV*NDFD were negatively correlated. Broderick et al. [54] reported that *IV*DMD and *IV*NDFD were negatively correlated. In the SWB group, elevating the excipient had no significant effect on *IV*DMD or *IV*NDFD. Gül et al. [27] found no significant effect of the addition of wheat bran on in vitro digestibility in the study on caramba silage. This is in agreement with the conclusion of the present experiment. *IV*DMD and *IV*NDFD levels can differ with silage

chemical composition [55]. The difference that led to this trend between the SMH and SWB groups may be related to the different NDF and ADF contents of the millet hull and wheat bran. *IV*OMD is used for the evaluation of available energy in feeds and can also be used in protein evaluation systems [56]. Previous studies have shown that the lignin content in feed is negatively correlated with *IV*OMD and *IV*DMD [57]. Lignin in crude fiber affects the digestion of cellulose and hemicellulose by rumen microorganisms. Thus, lignin affects the digestibility of roughage in ruminants [58]. In this study, *IV*DMD, *IV*OMD, and *IV*CPD decreased with decreasing moisture content in the SMH group. Rafael Monteiro Araújo et al. reported that the addition of coffee husks to maize silage reduced the *IV*DMD, *IV*OMD, *IV*CPD, and *IV*NDFD [59]. This may be because, for the SMH group, the loss of DM from the silage was reduced when the moisture content was increased to 65%, and the CP and residual WSC content of the feed was better utilized during rumen fermentation. As a result, the SMH group exhibited greater *IV*DMD and *IV*CPD at 65% moisture content ($T_3$ treatment).

The addition of LAB at 55% moisture content ($T_1$ treatment) significantly increased *IV*CPD in the SMH and SWB groups. This is likely because the addition of LAB inhibited protease activity and reduced the degradation of CP, increasing *IV*CPD. It has been shown that inoculation of corn silage with *Lactobacillus buchneri* and *Lactobacillus plantarum*, alone or in combination, had no significant effect on the *IV*NDFD, regardless of whether the moisture content of the silage was high or low [40]. Similarly, Filya reported that silage additives did not affect the *IV*DMD, *IV*OMD, or *IV*NDFD of wheat, sorghum, or corn silages [60]. This finding is similar to the results of this study. This is because LAB have the ability to produce acid and does not have the ability to produce enzymes. This explains why LAB has little effect on digestibility. The inconsistent results of the same silage additive on in vitro digestibility may be related to many factors, including the different harvesting periods of the raw material, the body conditions of the feeding fistulae, and the basal diet. In this experiment, with the same amount of LAB, the in vitro digestibility of SMH was lower than that of the SWB group. As shown in Table 1, the OM and CP contents of millet hull were lower than those of WB, whereas the NDF, ADF, and ADL contents were greater than those of wheat bran. This difference might explain why the in vitro digestibility of SWB was greater.

## 5. Conclusions

Overall, the silage constituent mixing proportion and LAB concentration affected the fermentation quality, chemical composition, and in vitro digestibility of the SCPBs silage. In this study, the addition of LAB improved fermentation quality and in vitro digestibility. The optimum mixing ratio for the SWB group was 9:1, and 8:2 for the SMH group. The addition of millet hull or wheat bran in combination with LAB is an effective way to promote SCPBs silage fermentation. However, we need to perform in vivo tests to further validate our study before it can be applied to large-scale production practices.

**Author Contributions:** Conceptualization, M.Y. and P.W.; methodology, M.Y.; software, M.Y. and J.D.; validation, M.Y. and F.L.; formal analysis, M.Y. and J.D.; investigation, Q.Y. and H.T.; resources, M.Y.; data curation, P.W. and T.Z.; writing—original draft preparation, M.Y. and Y.J.; writing—review and editing, P.W. and F.L.; visualization, P.W.; supervision, H.T.; project administration, P.W. and B.Y.; funding acquisition, P.W. and B.Y. All authors have read and agreed to the published version of the manuscript.

**Funding:** This research was supported by the Jilin Province Agricultural Key Core Technology Demonstration and Promotion (Industrial Technology System) Project (JARS-2024-0703) and funds from the China Agriculture Research System (CARS-37).

**Institutional Review Board Statement:** This study strictly followed the Chinese Laboratory Animal Welfare Ethical Review Guidelines and was approved by the Laboratory Animal Welfare Ethics Committee of Jilin University (permit number: SY202009600).

**Data Availability Statement:** Data are contained within the article.

**Acknowledgments:** We thank Zhihui Zhang and Tongyu Xinyu Beixian Rice Technology Co., Ltd. for providing the test material.

**Conflicts of Interest:** The authors declare no conflicts of interest.

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
