# Peer review of "Fermentation Quality and In Vitro Digestibility of Sweet Corn Processing Byproducts Silage Mixed with Millet Hull or Wheat Bran and Inoculated with a Lactic Acid Bacteria"

_fermentation, doi:10.3390/fermentation10050254_

Round 1
Reviewer 1 Report
Comments and Suggestions for Authors
The study addresses the use of ensiling the fresh by products which is a valuable thing to do for the ecosystem and the economies; the use of low-cost additives such as wheat bran and millet hull are both interesting and good for the reader to know their effect as well as the addition of lactic acid bacteria. The introduction is sufficient, and the methods are well described. The results presentation needs enhancements as for example in table 1 and all other tables, it would be much easier to read with full names rather than a list of abbreviations such as DM, OM, CP, etc. There are enough spaces to write these in full. Some table could also be replaced with figures for ease of interpretation.
Author Response
Dear reviewer, thank you for your encouraging and warm comments and suggestions, all of your suggestions are very important, and they all have important guiding significance for our future research work. Based on this we have revised and (we think) strengthened our paper.
Point 1: The results presentation needs enhancements as for example in table 1 and all other tables, it would be much easier to read with full names rather than a list of abbreviations such as DM, OM, CP, etc.
Response 1: We are very much in favour of your suggestion. We have reduced the number of abbreviations in the paper and revised the items of Table 1 and other tables.(Table 1,2 and 3)
Point 2: Some table could also be replaced with figures for ease of interpretation.
Response 2: We very much recognize this piece of your advice. We tried charting before, but due to the amount of data in this experiment. Making charts did not function well. Thank you very much for your suggestion, we will consider this in our subsequent experiments.
Thank you again for your suggestions and hope to learn more from you.
Reviewer 2 Report
Comments and Suggestions for Authors Notes on methods: Whether microbiological raw materials (sweet corn, millet hull, wheat bran) for silage production were tested before Lactobacillus plantarum LP1 was added. Were there any lactic acid bacteria (LAB) other than Lactobacillus plantarum LP1 in the raw material? Is the ensiling effect of the raw material the result of only Lactobacillus plantarum LP1 or also other LAB? Please present the results of the microbiological composition of the raw material and silage after fermentation?
Author Response
Dear reviewer, thank you for your encouraging and warm comments and suggestions, all of your suggestions are very important, and they all have important guiding significance for our future research work. Based on this we have revised and (we think) strengthened our paper.
Point 1: Whether microbiological raw materials (sweet corn, millet hull, wheat bran) for silage production were tested before Lactobacillus plantarum LP1 was added.
Response 1: We very much agree with your suggestion. We supplemented Table 1 with colony counts of raw materials, wheat bran and millet hull before silage. (Table 1)
Point 2: Were there any lactic acid bacteria (LAB) other than Lactobacillus plantarum LP1 in the raw material? Is the ensiling effect of the raw material the result of only Lactobacillus plantarum LP1 or also other LAB?
Response 2: We are very much in favour of your suggestion. Many other lactic acid bacteria are attached to the surface of the raw material. Currently, this question could not be answered in this experiment. We will explore this question in subsequent experiments using 16sRNA, metagenomics or qPCR techniques. In this experiment, we can only conclude that the difference between the LAB and control groups is due to the addition of LP1.
Point 3: Please present the results of the microbiological composition of the raw material and silage after fermentation?
Response 3 :We strongly endorse this suggestion. We supplemented Table 1 with colony-forming units of raw materials, wheat bran and millet hull before silage. We did not test for microbiological composition in this experiment. Thank you for your suggestion to bring this to our attention. In subsequent experiments, we will investigate the microbiological composition of raw materials and silage.(Table 1)
Thank you again for your suggestions and hope to learn more from you.
Reviewer 3 Report
Comments and Suggestions for Authors
The article examines the influence of lactic acid bacteria on ensiling. The results obtained suffer from poor presentation. The article is difficult to read and understand due to the vast number of abbreviations. Some of them are not even described, for example, "IVCPD" in the abstract.
Recommendations:
Reduce the number of abbreviations to a minimum, for example, "WB" and "MH" would not take up as much space to write, and would greatly facilitate reading. This also applies to most data in tables.
Do not start the abstract with preliminary experiments that are not the subject of this article.
The introduction should be expanded with the contents of silages, how their nutritional value is improved by fermentation, etc. How do lactic acid bacteria improve qualities, what genera and species have been used so far, and for what purpose?
In the tables, most abbreviations can be written in full, in this form the understanding of the results is difficult. How would you explain the small differences in IVCPD, IVGED, and IVNDFD shown in Table 3 for silage with and without LAB?
The role of lactic acid bacteria is central to this paper. To be able to enrich the discussion of the results, it is imperative to include (both in Methods and Results):
- the type of lactic acid bacteria, the origin of strains (collection), and the amount of live cells when inoculating the silage,
- monitoring the number of colony-forming units after 45 days and before the chemical analyses.
Comments on the Quality of English LanguageModerate editing of the English language is required.
Author Response
Dear reviewer, thank you for your encouraging and warm comments and suggestions, all of your suggestions are very important, and they all have important guiding significance for our future research work. Based on this we have revised and (we think) strengthened our paper.
Point 1: The article is difficult to read and understand due to the vast number of abbreviations. Some of them are not even described, for example, "IVCPD" in the abstract. Reduce the number of abbreviations to a minimum, for example, "WB" and "MH" would not take up as much space to write, and would greatly facilitate reading. This also applies to most data in tables.
Response 1: We very much agree with your suggestion. We reduced the number of abbreviations in the articles and modified the items in the tables. We have also modified the IVCPD in the abstract. (PDF:36)
Point 2:Do not start the abstract with preliminary experiments that are not the subject of this article.
Response 2: We are very much in favour of your suggestion. We have modified the abstract (PDF:17-23)
Point 3:The introduction should be expanded with the contents of silages, how their nutritional value is improved by fermentation, etc. How do lactic acid bacteria improve qualities, what genera and species have been used so far, and for what purpose?
Response 3: We very much recognize this piece of your advice. We added these to the introduction.(PDF:59-61;78-84)
Point 4:In the tables, most abbreviations can be written in full, in this form the understanding of the results is difficult. How would you explain the small differences in IVCPD, IVGED, and IVNDFD shown in Table 3 for silage with and without LAB?
Response 4: We strongly endorse this suggestion. We have reduced the abbreviations and modified the tables. To the best of our knowledge, most Lactobacillus additives are not enzyme-producing, only acid-producing. Therefore lactic acid bacteria have no fibre degrading ability. This explains why LAB have little effect on digestibility. In this experiment, the addition of LAB had a small effect on the CP content of the SMH and SWB groups. This resulted in a small effect of adding LAB on IVCPD. Filya reported that silage additive did not affect the IVDMD, IVOMD, or IVNDFD of wheat, sorghum, or corn silages. This is similar to the results of our experiment. (PDF:443-445, doi:10.1046/j.1365-2672.2003.02081.x.)
Point 5:The role of lactic acid bacteria is central to this paper. To be able to enrich the discussion of the results, it is imperative to include (both in Methods and Results)the type of lactic acid bacteria, the origin of strains (collection), and the amount of live cells when inoculating the silage.
Response 5: Thank you for your suggestions. We listed the species of lactic acid bacteria, the source of the strain (collection) and the number of viable cells at the time of silage inoculation in Material and Methods. (PDF:96-99)
Point 6:monitoring the number of colony-forming units after 45 days and before the chemical analyses.
Response 6: We very much recognize this piece of your advice. We supplemented Table 1 with colony-forming units of raw materials, wheat bran and millet hull before silage. However, we did not monitor the number of colony-forming units. Thank you for your suggestion again. We will take this into account in our subsequent experiments. (Table 1)
Thank you again for your suggestions and hope to learn more from you.
Round 2
Reviewer 3 Report
Comments and Suggestions for Authors
The authors have responded to all comments. They added the missing data and the necessary counting of the colonies of lactic acid bacteria, yeasts and possibly molds. They corrected the text in detail and answered the questions convincingly. The article may be published.
Comments on the Quality of English LanguageModerate editing of English language required.